# Nucleus-Independent Chemical Shift (NICS) as a Criterion for the Design of New Antifungal Benzofuranones

**DOI:** 10.3390/molecules26165078

**Published:** 2021-08-21

**Authors:** María de los Ángeles Zermeño-Macías, Marco Martín González-Chávez, Francisco Méndez, Arlette Richaud, Rodolfo González-Chávez, Luis Enrique Ojeda-Fuentes, Perla del Carmen Niño-Moreno, Roberto Martínez

**Affiliations:** 1Facultad de Ciencias Químicas, Universidad Autónoma de San Luis Potosí, Av. Dr. Manuel Nava No. 6 Zona Universitaria, San Luis Potosí 78210, Mexico; angeles.zermeno@uaslp.mx (M.d.l.Á.Z.-M.); rodolfo.gonzalez@uaslp.mx (R.G.-C.); luisenrique.ojedaf@yahoo.com.mx (L.E.O.-F.); 2Departamento de Química, División de Ciencias Biológicas e Ingeniería, Universidad Autónoma Metropolitana, Unidad Iztapalapa, Ciudad de México 09340, Mexico; avrt@xanum.uam.mx; 3CEMHTI-CNRS, UPR3079, Site Haute Température, CS 90055, 1D avenue de la Recherche Scientifique, CEDEX 2, 45071 Orléans, France; 4Centro de Investigación en Ciencias de la Salud y Biomedicina (CICSaB), UASLP. Av. Paseo de los Derechos Humanos No. 300, Lomas de San Luis, San Luis Potosí 78210, Mexico; ncarmenp@uaslp.mx; 5Instituto de Química, Universidad Nacional Autónoma de México, Circuito Exterior de Ciudad Universitaria, Ciudad de México 04510, Mexico; robmar@servidor.unam.mx

**Keywords:** NICS, aromaticity, benzofuran-4-one, antifungal activity, quantitative structure–activity relationship

## Abstract

The assertion made by Wu et al. that aromaticity may have considerable implications for molecular design motivated us to use nucleus-independent chemical shifts (NICS) as an aromaticity criterion to evaluate the antifungal activity of two series of indol-4-ones. A linear regression analysis of NICS and antifungal activity showed that both tested variables were significantly related (*p* < 0.05); when aromaticity increased, the antifungal activity decreased for series I and increased for series II. To verify the validity of the obtained equations, a new set of 44 benzofuran-4-ones was designed by replacing the nitrogen atom of the five-membered ring with oxygen in indol-4-ones. The NICS(0) and NICS(1) of benzofuran-4-ones were calculated and used to predict their biological activities using the previous equations. A set of 10 benzofuran-4-ones was synthesized and tested in eight human pathogenic fungi, showing the validity of the equations. The minimum inhibitory concentration (MIC) in yeasts was 31.25 µg·mL^–1^ for *Candida glabrata*, *Candida krusei* and *Candida guilliermondii* with compounds **15**-**32**, **15**-**15** and **15**-**1**. The MIC for filamentous fungi was 1.95 µg·mL^–1^ for *Aspergillus niger* for compounds **15**-**1**, **15**-**33** and **15**-**34****.** The results obtained support the use of NICS in the molecular design of compounds with antifungal activity.

## 1. Introduction

Aromaticity began as a descriptor of the special stability of benzene. In 1933, Pauling and Wheland found that the resonance between the Kekulé (the most important), Dewar, Claus and Armstrong–Baeyer structures imparts to benzene its peculiar aromatic properties [1]. In 1993, Schleyer et al. developed nucleus-independent chemical shifts (NICS) as an aromaticity criterion [2]. The NICS have been widely used in the study of annulenes, polycyclic aromatic hydrocarbons, hydrocarbon pericyclic reaction transition states, σ-aromaticity/σ-antiaromaticity, Möbius aromaticity [3], the relationship between bond dissociation enthalpy and the antioxidant activity [4].

In 2014, Wu et al. suggested that aromaticity may have considerable implications for molecular design [5]. They investigated hydrogen bonding–aromaticity relationships for different H-bonded substrates and concluded that “such relationships may have considerable implications for molecular design, as the functions of many heterocyclic biomolecules and drugs rely on the binding via H-bonds of a dominant keto/imine or enol/amine tautomer” [5]. The proposal should be very useful for the synthesis of biologically active compounds that have aromatic moieties in their structure, for example, the biologically active aromatic compounds that constitute an important group of drugs widely used in various medicinal applications [6]. However, to the best of our knowledge, there is no evidence for the use of NICS in the experimental design of aromatic compounds with biological activity, especially with antifungal properties.

Recently, antifungal activity has acquired great importance due to the increasing rates of occurrence and mortality as a result of opportunistic mycosis caused by species of *Aspergillus*, *Candida*, *Coccidioides*, *Cryptococcus* and *Histoplasma* [7]. Different kinds of antifungal compounds have been used as therapeutic options for the treatment of systemic fungal infections. Those drugs have disadvantages regarding high toxicity, different activity spectrums, safety in their administration and a variety of pharmacokinetic properties. Actually, some etiological agents have developed resistance by different mechanisms of action [8,9,10], reinforcing the studies to develop new antifungal agents [11,12,13,14,15,16,17,18,19,20,21].

As part of our continuing interest in the design of non-coordinating indolones as potential inhibitors for lanosterol 14-α-demethylase (CYP51), we synthesized two series of indol-4-ones that showed antifungal activity against *Aspergillus* (filamentous) and *Candida* (yeast) fungi (Figure 1). The docking study suggested that the antifungal activity of indol-4-one increases when its keto moiety increases the capacity to form a hydrogen bond with the Arg96 residue of *Mycobacterium tuberculosis* CYP51 [22]. Molecular electrostatic potential surfaces and Fukui functions suggested that there is a correlation between the antifungal activity and the electrophilic/nucleophilic character of the keto/pyrrole moieties and the polarity of indol-4-ones [23]. In this work, we calculated the NICS(0) and NICS(1) of indol-4-ones; a linear regression analysis showed that when aromaticity increased, the antifungal activity decreased for series I and increased for series II. As a way to use the obtained equations and test the statement of Wu, Jackson and Schleyer, we designed a new set of 44 benzofuran-4-ones by replacing the pyrrole ring with furan in indol-4-ones (replacing the nitrogen atom with oxygen in the five-membered ring). The 10 most promising molecules were synthesized and tested against eight human pathogenic fungi. The statistical analysis of the NICS values and the antifungal activity of the 10 benzofuran-4-ones followed the expected trend, showing the validity of the statement of Wu et al.

## 2. Results and Discussion

### 2.1. NICS(0) and NICS(1) of Indol-4-ones

Table 1 shows the calculated aromaticity values for the five-membered ring (pyrrole ring) of the set of 15 indol-4-ones. The trends of the NICS(0) values for series I and II are **6** > **7f** > **7c** > **7g** > **7d** > **7e** > **7a** > **7b** and **8g** > **8e** > **8d** > **8c** > **8f** > **8a** > **8b**, while the trends of the NICS(1) values for series I and II are **6** > **7f** > **7c** > **7d** = **7b** > **7e** > **7a** > **7g**, and **8g** > **8e** > **8c** > **8d** > **8f** > **8b** > **8a**. We can observe for the unsubstituted molecule 6 that the replacement of the H atom in the NH or CH_3_ moiety by a phenyl group (**7a** or **8a**) decreases its aromaticity (except for **8e** and **8g**). The additional substitution of the H atom in the phenyl ring of 7a and 8a by X causes the aromaticity of series I (**7a**–**7g**) to be less than that of series II (**8a**–**8g**). The substitution of the H atom in the phenyl ring with fluorine at position 2 (**7b** and **8b**) greatly reduces the aromaticity (NICS(0)) while the effect of substitution with chlorine in position 2 (**8e**) and disubstitution in positions 2 and 4 (**8g**) increases the aromaticity.

### 2.2. Structure–Activity Relationship: Aromaticity vs. Antifungal Activity

Simple linear regression analysis of the antifungal activity of the two series of 15 indol-4-ones and their NICS(0) and NICS(1) values was conducted. The antifungal activity measured as the minimum inhibitory concentration (MIC) values against *Aspergillus* (filamentous) and *Candida* (yeast) fungi were taken from our previous report [22]. Table 2 shows the Pearson (*p*) and sample (*r*) correlation coefficients. For series I, good correlation values were obtained (*p* < 0.05) for the filamentous fungi *Aspergillus niger* (evaluated at 48 h): NICS(0), *p* = 0.002, *r* = 0.94; NICS(1), *p* = 0.001, *r* = 0.95; *Aspergillus niger* (evaluated at 72 h): NICS(0), *p* = 0.015, *r* = 0.89; NICS(1), *p* = 0.068, *r* = 0.78, *Aspergillus fumigatus* (evaluated at 48 h): NICS(0), *p* = 0.024, *r* = 0.77; and As*pergillus fumigatus* (evaluated at 72 h): NICS(0), *p* = 0.004, *r* = 0.88, NICS(1), *p* = 0.002, *r* = 0.91. We observed a major biological activity for filamentous fungi with compounds of series I [22]. Series I showed an inverse proportional relationship between both variables (see Appendix A): the antifungal activity increased when the aromaticity of the pyrrole ring decreased.

For series II (see Table 3), the statically significant values were obtained for the yeast fungi *Candida albicans* (evaluated at 48 h): NICS(0), *p* = 0.050, *r* = 0.75; NICS(1), *p* = 0.021, *r* = 0.83; *Candida glabrata* (evaluated at 24 h): NICS(0), *p* = 0.050, *r* = 0.75; NICS(1), *p* = 0.021, *r* = 0.83; *Candida tropicalis* (evaluated at 48 h): NICS(1), *p* = 0.029, *r* = 0.81; and *Candida parapsilosis* (evaluated at 24 h): NICS(0), *p* = 0.004, *r* = 0.91, NICS(1), *p* = 0.025, *r* = 0.82. We observed major biological activity for yeasts with compounds of series II [22]. Series II showed a direct proportional relationship between both variables (see Appendix A): the antifungal activity increased when the aromaticity of the pyrrole ring increased.

### 2.3. Design of New Compounds

The linear regression analysis showed the importance of the pyrrole ring (five-membered ring) aromaticity in the activity of the indol-4-one compounds. Two significant resonance structures of indol-4-ones should be involved in the aromaticity of the pyrrole ring and biological activity (Scheme 1 and Scheme 2). The keto and enolate resonance structures are obtained by displacing the lone pair of the nitrogen atom in/out of the pyrrole ring. The aromaticity of the pyrrole ring decreases from the keto to the enolate structures because the electron density is displaced from the pyrrole ring to the oxygen atom. The antifungal activity of the indol-4-one compounds is related with the enolate resonance structure for series I and the keto resonance structure for series II.

To extrapolate the previous linear regression equations for aromaticity and antifungal activity, we made a bioisosteric change in the five-membered ring of indol-4-ones substituting the nitrogen atom with an oxygen atom to obtain benzofuran-4-ones. Replacement of the pyrrole ring with furan decreases the aromaticity of indol-4-ones since the furan ring is less aromatic than the pyrrole ring [24]. Therefore, a new set of benzofuran-4-ones (**14**, **15**-**1** to **15**-**43**) was designed and the NICS(0) and NICS(1) were calculated, and as we expected, benzofuran-4-ones had lower aromaticity values than indol-4-ones. Table 4 shows that the NICS(0) and NICS(1) values ranged from −7.82 to −8.97 ppm and from –5.62 to –7.92 ppm, respectively. We observed for the unsubstituted benzofuran-4-one 14 that the replacement of the H atom in the CH_3_ moiety by a phenyl group (**15**-**1**) decreased its aromaticity. The additional substitution of the H atom in the phenyl ring of 15-1 with R caused the aromaticity (measured in terms of NICS(0)) to decrease for R = 4-OCF_3_ (**15**-**25**) and increase for R = 2-CF_3_ (**15**-**35**), while for NICS(1), the aromaticity decreased for R = 4-Phe (**15**-**28**) and increased for R = 2-NO_2_ (**15**-**28**). Therefore, the aromaticity for benzofuran-4-ones decreased and increased with electron-donating and electron-withdrawing substituents.

Once the antifungal activity of benzofuran-4-ones was interpolated using the equations obtained in the linear regression analysis with indol-4-ones (Appendix A) and the NICS(0) and NICS(1) values (Table 4), some of the proposed benzofuran-4-ones were synthesized to corroborate the theoretical prediction. The criteria used to select the benzofuran-4-ones to be synthesized were as follows: (a) series of compounds that could exhibit high, intermediate and low calculated antifungal activity based on the interpolation; (b) compounds that influenced the calculated activity of different yeasts or filamentous fungi; (c) compounds the chemical structure whereof allows obtaining benzofuran-4-ones with a variety of substituents in the phenyl ring: strong/weak activators/deactivators; (d) molecules with the same groups in identical positions as in indol-4-ones for later comparisons; (e) easy access to reagents and the synthetic procedure. The benzofuran-4-ones selected were as follows: **14**, **15**-**1**, **15**-**3**, **15**-**11**, **15**-**15**, **15**-**28**, **15**-**32**, **15**-**33**, **15**-**34** and **15**-**41**.

### 2.4. Synthesis of Benzofuran-4-ones

The synthesis of benzofuran-4-one 14 (see Scheme 3) began with preparation of the tricarbonyl compound 12 from dimedone 11 via nucleophilic substitution by chloroacetone (using EtONa as the base) under the N_2_ atmosphere (enolate formation) [22]. Intramolecular cyclization of **12** through the Paal–Knorr reaction was performed using the methodology reported by Goncalves et al. using trimethylsilyl chloride under microwave irradiation [25]. For compounds **15**-**1**, **15**-**3**, **15**-**11**, **15**-**15**, **15**-**28**, **15**-**32**, **15**-**33**, **15**-**34** and **15**-**41**, bromoacetophenones **10**-**1**, **10**-**3**, **10**-**11**, **10**-**15**, **10**-**28**, **10**-**32**, **10**-**33**, **10**-**34** and **10**-**41** were synthetized from the R-acetophenones **9**-**1**, **9**-**3**, **9**-**11**, **9**-**15**, **9**-**28**, **9**-**32**, **9**-**33**, **9**-**34** and **9**-**41** using *p*-toluensulfonic acid and *N*-bromosuccinimide (NBS). Nucleophilic substitution with the enolate of dimedone 11 (using K_2_CO_3_ as the base) yielded the tricarbonyl compounds **13**-**1**, **13**-**3**, **13**-**11**, **13**-**15**, **13**-**28**, **13**-**32**, **13**-**33**, **13**-**34** and **13**-**41**. Benzofuran-4-ones **15**-**1**, **15**-**3**, **15**-**11**, **15**-**15**, **15**-**28**, **15**-**32**, **15**-**33**, **15**-**34** and **15**-**41** were obtained via Paal–Knorr reactions using trimethylsilyl chloride under microwave irradiation [25]. This methodology for obtaining furanes is free from the use of strong acids and toxic solvents and the reaction time is short; the products precipitate and generally do not require purification by chromatographic methods, which indicates tha the reaction is efficient and could be considered within the concept of green chemistry.

**Reaction conditions: (****i)** 1 eq NBS/1 eq TsOH H_2_O/ACN/reflux; **(ii)** 1 eq ClCH_2_COCH_3_/1 eq EtONa/EtOH/N_2_ atm.; **(iii)** 1.8 eq K_2_CO_3_/CHCl_3_/12 h, N_2_ atm.; **(iv)** 1 eq (CH_3_)_3_SiCl/CH_3_OH/MW, 90°C, 8 min, 250 W.

### 2.5. Antifungal Activity of Benzofurans

Table 5 shows the in vitro antifungal activities of compounds **14** and **15**. They were determined using six strains of yeast (*C. albicans*, *C. glabrata*, *C. krusei*, *C. tropicalis*, *C. guilliermondii* and *C. parapsilosis*) and two strains of fungi (*A. fumigatus* and *A. niger*). The MIC values of fluconazole and itraconazole are shown as reference antifungal drugs. The lowest and highest MIC of the yeasts were 31.25 µg·mL^–1^ and >500 µg·mL^–1^. For filamentous fungi, the MIC values fluctuated between 1.95 µg·mL^–1^ and >500 µg·mL^–1^. *C. glabrata*, *C. krusei* and *C. guilliermondii* tested at 24 h presented the MIC of 31.25 µg·mL^–1^ for compounds **15**-**32**, **15**-**15** and **15**-**1**. The previous compounds had phenyl substituents acting as weak deactivators, strong activators and no substituted phenyl ring. *A. niger* tested at 48 h and 72 h had similar MIC values of 1.95 µg·mL^–1^ for compounds **15**-**1**, **15**-**33** and **15**-**34**, compounds without a substituent and weak deactivators, respectively. The yeasts presented decreased antifungal activity (high MIC values) concerning fungi. In Table 6, we show the summary of NICS and MIC values of indol-4-ones and benzofuran-4-ones with similar structures. Four compounds from each group had comparable moieties: **6** vs. **14**, **8a** vs. **15**-**1**, 8c vs. **15**-**33** and 8d vs. **15**-**34**. The compound that decreased the MIC values in more species was 15-1 (four yeasts and two filamentous fungi).

The Mann–Whitey *U* test was performed to evaluate the difference in the antifungal activity of two different kinds of compounds: indol-4-ones and benzofuran-4-ones with the same substituent (**6**, **8a**, **8c**, **8d**, **14**, **15**-**1**, **15**-**33** and **15**-**34**). The summary of the statistical results is shown in Table S-3. We found a significant variation in the antifungal activity for yeasts (24 h of testing) and filamentous fungi (48 and 72 h of testing) (*p* < 0.05). No significant difference was found between the antifungal activity of benzofuran-4-ones and indol-4-ones in the yeasts evaluated at 48 h (*p >* 0.05). For the yeasts at 24 h, benzofuran-4-ones showed less biological activity in contrast to indol-4-ones (mean ranks: 33.92 and 15.08, respectively). Therefore, benzofuran-4-ones had higher antifungal activity (evaluated at 48 and 72 h) in filamentous fungi than indol-4-ones (mean ranks: 5.50 and 11.50) and there was no improvement in the biological activity of any yeast.

### 2.6. Statistical Analysis between the Calculated and Experimental MIC Values

Statistical comparison between the calculated and experimental MIC values was performed using the Mann–Whitney *U* test only with comparable molecules: **14**, **15**-**1**, **15**-**33** and **15**-**34**. The NICS(0) and NICS(1) of benzofuran-4-ones were calculated and substituted in the equations of indol-4-ones. The summary of the statistical analysis is shown in Table S-4. Even though the equations used to evaluate the calculated antifungal activity were obtained from indole-4-one compounds, the equations fit well to predict the activity of benzofuran-4-ones. The calculated and experimental MIC values were similar in general. The results obtained support the use of NICS in the molecular design of compounds with antifungal activity.

## 3. Materials and Methods

### 3.1. Computational Details

The geometries of the molecules **6**, **7a**-**g**, **8a**-**g** [5], **14** and **15**-**1** to **15**-**43** were fully optimized at the gas phase and at the B3LYP/6-311 + G(*d,p*) level of theory [26,27]. Scheme 4 shows the optimized geometries of the main compounds. The B3LYP functional was proved to be valid previously in the calculation of NICS of pyrrole and pyridine [28], the chemical reactivity of oxazole [29] and imidazole [30] and the basicity of phenol [31]. The calculations were performed using the Gaussian 09 program package [32]. Minimum energy structures were obtained and verified by all positive frequencies for each molecule.

The NICS(0) and NICS(1) values were obtained from the isotropic chemical shift revealed by the gauge-independent atomic orbital (GIAO) calculations of the geometric center at the five-membered ring of each structure at and 1 Å above the molecular plane, respectively [2,3]. Scheme 5 shows the ghost atoms locations to calculate NICS(0) and NICS(1).

### 3.2. Structure–Activity Relationship (SAR) Statistical Procedure

The MIC values of a set of indol-4-ones versus NICS(0) and NICS(1) values at 24 and 48 h for yeasts and 48 and 72 h for filamentous fungi were plotted (Appendix A). Simple regression analysis for each plot was performed and Pearson and significant one coefficients were obtained using the SAS software [33]. The *p* < 0.05 value was considered significant for this study. The best equations and the classification of the indol-4-ones used by us were used to predict the antifungal activity of the new compounds.

### 3.3. Synthesis of Benzofuran-4-ones

All the reagents and solvents used were reagent grade (Sigma-Aldrich Co. St. Louis, MO, USA). Monitoring of reactions was performed by means of thin-layer chromatography (Sigma-Aldrich Inc. St. Luis, MO, USA) using aluminum and silica gel 60 support chromatofolios, with an F_254_ indicator from Merck.

A medium-resolution column chromatography column was used to separate the reaction mixtures using silica gel as the stationary phase and a mixture of hexane ethyl acetate as the mobile phase. Synthesis intermediaries and final products were characterized by means of infrared spectroscopy with the Thermo Nicolet Nexus 470 FT-IT E.S.P. equipment (Thermo Nicolet Co. Madison, WI. USA), ^1^H nuclear magnetic resonance (^1^H-NMR) and ^13^C nuclear magnetic resonance (^13^C-NMR) with the Variant Gemini 200 MHz and 50 MHz, respectively) (Varian associates, Palo Alto, CA. USA) and Jeol (300 MHz and 70 MHz, respectively) equipment (Jeol USA, Inc, Peabody, MA. USA) utilizing dimethylsulfoxide (DMSO-d_6_) as the solvent. Chemical shifts are expressed with δ ppm values relative to tetramethylsilane (TMS) used as the internal standard (s = singlet, d = doublet, t = triplet, q = quadruplet, sext = sextet and m = multiplet). Coupling constants (J) are shown in Hertz (Hz).

5,5-dimethyl-2-(2-oxopropyl)cyclohexane-1,3-dione (**12**)

A mixture of 14 mmol of sodium ethoxide, 2 g (14 mmol) of 5,5-dimethylciclohexane-1,3-dione and 1.33 mL (14 mmol) of chloroacetone in 20 mL of ethanol was refluxed for 30 min and then letting cold. Sodium chloride was separated by filtration and their filtrate was vacuum-concentrated. The residual oil was dissolved in a 1:1 mixture of chloroform (20 mL) and 10% sodium hydroxide (20 mL). The aqueous phase was separated and then extracted with chloroform (20 mL); this aqueous phase was cooled in an ice bath, acidified with chloride acid, and the organic phase was extracted with chloroform (3 × 20 mL). The organic extracts were dried with anhydrous sodium sulfate and filtered, the resulting solution was concentrated by evaporation in vacuum. Recrystallization was performed in acetone, and compound 12 was obtained as colorless crystals, yield: 80%, mp: 133–135 °C. ^1^H-NMR (200 MHz, DMSO-d_6_*) δ* = 0.99 (s, 6H), 1.98 (s, 3H), 2.22 (s, 4H), 3.16 (s, 1H), 3.33 (s, 1H), 10.62 (s, 1H). EI-MS: *m*/*z* (%): 196 (M+, 51), 154 (89), 125 (24), 98 (100), 55 (45). IR (KBr): λ = 2962, 2931, 1720, 1645, 1155, 1064 cm^−1^.

General procedure for the synthesis of 2-(2-(R)-2-oxoethyl)-5,5-dimethylcyclohexane-1,3-diones (13s)

A suspension with dimedone 11 (1 eq), bromoacetophenones 10 (1 eq) and anhydrous K_2_CO_3_ (1.8 eq) in chloroform was agitated to room temperature for 12 h. Then, the mixture was filtered. The salts were dissolved in water and the filtered solutions were acidulated with concentrated HCl. The precipitate was separated by filtration and washed with water.

5,5-dimethyl-2-(2-oxo-2-phenylethyl)cyclohexane-1,3-dione (**13**-**1**)

White dust, yield: 39.5%, m.p.: 136–138 °C. ^1^H-NMR (200 MHz, DMSO-d_6_) δ = 0.97 (s, 6H), 2.10 (s, 4H), 3.79 (s, 2H), 7.44–7.63 (m, 3H), 7.90–7.95 (m, 2H), 10.96 (s, 1H). EI-MS: *m*/*z* (%): 258 (M+, 11), 140 (30), 105 (34), 83 (100), 54 (55). IR (KBr): λ = 2960, 2869, 2680, 2630, 2576, 2532, 1617, 1577, 1529, 1471, 1349, 14305, 1226, 1145 cm^−1^.

2-(2-(4-(1H-pyrrol-1-yl)phenyl)-2-oxoethyl)-5,5-dimethylcyclohexane-1,3-dione (**13**-**3**)

Brown dust, yield: 54%, m.p.: 60–62 °C. ^13^C-NMR (101 MHz, DMSO) δ = 190.15, 179.58, 176.08, 175.18, 99.30, 49.99, 33.33, 32.71, 31.94, 29.55, 28.97. IR (KBr): λ = 2952, 2670, 1683, 1609, 1374, 1243, 1204, 1039, 764 cm^−1^.

2-(2-(2-methoxyphenyl)-2-oxoethyl)-5,5-dimethylcyclohexane-1,3-dione (**13**-**11**)

White dust, yield: 65%, m.p.: 151 °C. ^1^H-NMR (300 MHz, DMSO) δ = 7.51 (d, *J* = 1.8 Hz, 1H), 7.47 (d, *J* = 7.5 Hz, 1H), 7.12 (d, *J* = 7.9 Hz, 1H), 7.02–6.94 (m, 1H), 3.37 (s, 3H), 2.20 (d, 4H), 0.98 (d, 6H). ^13^C-NMR (101 MHz, DMSO) δ = 199.40, 157.89, 132.96, 129.51, 128.42, 120.23, 112.11, 108.65, 102.44, 55.67, 37.22, 32.13, 27.93. TOFMS: *m/z* (%): 289 (M+, 99.2), 271 (0.8), 181 (7.5), 152 (0.8), 135 (1.5). IR (KBr): λ = 3178, 2958, 2663, 1666, 1594, 2595, 1467, 1394, 1255, 1201, 1155, 1045, 761 cm^−1^.

2-(2-(2,5-(dimethoxyphenyl)-2-oxoethyl)-5,5-dimethylcyclohexane-1,3-dione (**13**-**15**)

White dust, yield: 51%, m.p.: 143–145 °C. ^1^H-NMR (400 MHz, DMSO) δ = 7.07 (d, *J* = 1.8 Hz, 2H), 7.03–7.00 (m, 1H), 3.81 (s, 3H), 3.36 (s, 6H), 2.21 (s, 4H), 0.98 (s, 6H). ^13^C-NMR (101 MHz, DMSO) δ = 198.86, 152.78, 152.25, 128.73, 118.64, 113.74, 108.60, 56.20, 55.50, 37.16, 31.76, 27.90. IR (KBr): λ = 3161, 2952, 2835, 2652, 1661, 1504, 1491, 1390, 1330, 1278, 1248, 1217, 1050, 817, 752 cm^−1^.

2-(2-([1,1′-biphenyl]-4-yl)-2-oxoethyl)-5,5-dimethylcyclohexane-1,3-dione (**13**-**28**)

Yellow dust, yield: 48%, m.p.: 158–160 °C. EI-MS: *m*/*z* (%): 334 (M+, 5.6), 319 (0.9), 181 (99.9), 152 (27.1), 151 (3.7), 55 (4.7), 55 (12.2), 41 (13.1), 29 (3.7). IR (KBr): λ = 2952, 2670, 1683, 1609, 1374, 1243, 1204, 1039, 764 cm^−1^.

2-(2-(4-chloro-3-nitrophenyl)-2-oxoethyl)-5,5-dimethylcyclohexane-1,3-dione (**13**-**32**)

White dust, yield: 66%, m.p.: 171–173 °C. ^1^H-NMR (400 MHz, DMSO) δ = 10.81 (s, 1H), 8.50 (d, *J* = 2.0 Hz, 1H), 8.18 (dd, *J* = 8.4, 2.1 Hz, 1H), 7.94–7.90 (m, 1H), 3.82 (s, 2H), 2.25 (s, 4H), 0.99 (s, 6H). ^13^C-NMR (101 MHz, DMSO) δ = 195.51, 147.70, 136.59, 132.51, 132.13, 128.94, 124.68, 107.69, 79.26–79.07, 32.89, 31.86, 27.82. IR (KBr): λ = 3091, 2952, 2926, 2656, 1696, 1596, 1560, 1534, 1387, 1356, 1226, 821 cm^−1^.

2-(2-(4-fluorophenyl)-2-oxoethyl)-5,5-dimethylcyclohexane-1,3-dione (**13**-**33**)

White dust, yield: 59%, m.p.: 96 °C. ^1^H-NMR (400 MHz, DMSO) δ = 8.05–7.99 (m, 2H), 7.36–7.28 (m, 2H), 3.34 (s, 3H), 2.46–2.04 (m, 4H), 1.02 (s, 6H). ^13^C-NMR (101 MHz, DMSO) δ = 196.05, 165.99, 163.49, 133.66, 130.71, 115.63, 115.41, 108.03, 32.47, 31.86, 27.89. IR (KBr): λ = 2960, 2923, 1670, 1597, 1385, 1255, 1223, 1165, 840, 671 cm^−1^.

2-[2-(2,4-difluorophenyl)-2-oxoethyl]-5,5-dimethylcyclohexane-1,3-dione (**13**-**34**)

Yellow dust, yield: 73%, m.p.: 138 °C. ^1^H-NMR (200 MHz, DMSO-d_6_) δ = 0.97 (s, 6H), 2.22 (s, 4H), 3.68 (d, 2H, *J* = 2.4 Hz), 7.18 (ddd, 1H, *J* = 7.2, 9.6 and 0.8 Hz), 7.37 (ddd, 1H, *J* = 11.45, 9.3 and 2.6 Hz), 7.81 (ddd, 1H, *J* = 16, 7.6 and 1.8 Hz), 10.61 (s, 1H). ^13^C-NMR (50 MHz, DMSO-d6) δ = 27.8, 31.8, 36.3, 44.6, 104.9, 107.7, 112.1, 122.7, 132.2, 158.8, 162.0, 163.9, 167.03, 172.9, 194.7. EI-MS: *m*/*z* (%): 294 (M +, 46), 153 (30), 141(100), 97 (24), 83 (14), 55 (8). IR (KBr): λ = 3077, 2964, 2892, 2657, 1695, 1608, 1577, 1243, 1199, 1147, 1099, 1039 cm^−1^.

5,5-dimethyl-2-(2-(4-nitrophenyl)-2-oxoethyl)cyclohexane-1,3-dione (**13**-**41**)

Orange liquid, yield: 28%. ^1^H-NMR (400 MHz, DMSO) δ = 8.34–8.29 (m, 2H), 8.17–8.11 (m, 2H), 3.41 (t, *J* = 103.4 Hz, 3H), 2.25 (s, 4H), 1.00 (s, 6H). TOFMS: *m*/*z* (%): 304 (M+, 99.2), 290 (0.8), 155 (0.8). ^13^C-NMR (101 MHz, DMSO) δ = 196.98, 144.11, 139.01, 135.78, 129.08, 128.41, 126.85, 108.16, 32.51, 31.88, 27.92. IR (KBr): λ = 2970, 2939, 1717, 1578, 1382, 1356, 1256, 1217, 1156, 1064, 608 cm^−1^.

General procedure for synthesis of compounds 14 and 15s

Trimethylsilyl chloride (1 eq) was added to a suspension with carbonyl compounds (4 eq, 50 mg) dissolved in MeOH (300 uL). The resulting suspension was collocated in a microwave oven with the following conditions: 90 °C, 250 W, 290 psi, 8 min [25]. The mixture was left to cool to room temperature and the solid was filtered and washed with hexane. The resultant mixture was purified with column chromatography using silica gel in a gradient of 3–20% AcOE:hexane as the eluent. The compounds were obtained after vacuum fraction evaporation.

2,6,6-trimethyl-6,7-dihydrobenzofuran-4(5H)-one (**14**)

Yellow liquid, yield: 72%. ^1^H-NMR (200 MHz, DMSO-d_6_) δ = 1.06 (s, 6 H), 2.22 (s, 3 H), 2.27 (s, 2 H), 2.62 (s, 2 H), 6.16 (s, 1 H). EI-MS: *m*/*z* (%): 178 (M+, 65), 121 (92), 93 (100), 66 (7), 42 (13). IR (KBr): λ = 2961, 2874, 1680, 1605, 1584, 1433, 1387, 1346, 1302, 1248, 1227, 1184, 1115, 1033, 930 cm^−1^.

6,6-dimethyl-2-phenyl-6,7-dihydrobenzofuran-4(5H)-one (**15**-**1**)

Colorless crystals, yield: 67%, m.p.: 103–105 °C. ^1^H-NMR (200 MHz, DMSO-d_6_) δ = 1.15 (s, 6H), 2.37 (s, 2H), 2.74 (s, 2H), 6.77 (s. 1H), 7.15–7.7 (m, 5H). EI-MS: *m*/*z* (%): 240 (M+, 62), 185 (31), 184 (88), 157 (10), 156 (75), 128 (13), 105 (100), 77 (28), 51 (9), 43 (4), 41 (8). IR (KBr): λ = 2950, 2870, 1665, 1440, 1220, 755, 680 cm^−1^.

2-(4-(1H-imidazol-1-yl)phenyl)-6,6-dimethyl-6,7-dihydrobenzofuran-4(5H)-one (**15**-**3**)

Light brown dust, yield: 50%, m.p.: 88–90 °C. IR (KBr): λ = 2956, 2927, 1657, 1609, 1375, 1226, 1153, 1016, 825 cm^−1^.

2-(2-metoxyphenyl)-6,6-dimethyl-6,7-dihydrobenzofuran-4(5H)-one (**15**-**11**)

White dust, yield: 57%, m.p.: 106–108 °C. ^1^H-NMR (400 MHz, DMSO) δ = 7.74 (dd, *J* = 7.8, 1.7 Hz, 1H), 7.33 (ddd, *J* = 8.9, 7.4, 1.7 Hz, 1H), 7.14 (d, *J* = 7.8 Hz, 1H), 7.05 (td, *J* = 7.7, 1.0 Hz, 1H), 7.01 (s, 1H), 2.87 (s, 2H), 2.36 (s, 2H), 1.09 (s, 6H). ^13^C-NMR (101 MHz, DMSO) δ = 193.17, 165.27, 155.16, 150.07, 129.18, 125.24, 121.01, 120.63, 117.74, 111.74, 105.13, 55.56, 51.37, 36.34, 34.94, 27.98. IR (KBr): λ = 3149, 2956, 2870, 1685, 1595, 1491, 1442, 1253, 1118, 1012, 752, 671 cm^−1^.

2-(2,5-dimethoxyphenyl)-6,6-dimethyl-6,7-dihydrobenzofuran-4(5H)-one (**15**-**15**)

Beige dust, yield: 56%, m.p.: 124–125 °C. ^1^H-NMR (400 MHz, DMSO) δ = 7.25 (d, *J* = 3.1 Hz, 1H), 7.08 (d, *J* = 9.0 Hz, 1H), 7.03 (s, 1H), 6.91 (dd, *J* = 9.0, 3.1 Hz, 1H), 3.87 (s, 3H), 3.76 (s, 3H), 2.88 (s, 2H), 2.36 (s, 2H), 1.09 (s, 6H). ^13^C-NMR (101 MHz, DMSO) δ = 193.12, 165.34, 153.10, 149.85, 149.48, 121.05, 118.28, 114.15, 112.95, 110.39, 105.47, 55.91, 55.52, 36.35, 34.94, 30.67, 27.97. IR (KBr): λ = 2962, 2871, 2643, 1685, 1611, 1446, 1403, 1282, 1240, 1049, 1110, 1018, 800, 733 cm^−1^.

2-([1,1′-biphenyl]-4-yl)-6,6-dimethyl-6,7-dihydrobenzofuran-4(5H)-one (**15**-**28**)

Beige crystals, yield: 83%, m.p.: 176 °C. ^1^H-NMR (400 MHz, DMSO) δ = 7.86–7.82 (m, 2H), 7.78–7.70 (m, 4H), 7.51–7.45 (m, 2H), 7.41–7.35 (m, 1H), 7.23 (s, 1H), 2.91 (s, 2H), 2.39 (s, 2H), 1.11 (s, 6H). IR (KBr): λ = 2952, 1670, 1448, 1396, 1213, 1117, 761 cm^−1^.

2-(4-chloro-3-nitrophenyl)-6,6-dimethyl-6,7-dihydrobenzofuran-4(5H)-one (**15**-**32**)

Light yellow crystals, yield: 85%, m.p.: 198–199 °C. ^1^H-NMR (400 MHz, DMSO) δ = 8.40 (d, *J* = 2.1 Hz, 1H), 8.05 (dd, *J* = 8.5, 2.2 Hz, 1H), 7.85 (d, *J* = 8.5 Hz, 1H), 7.48 (s, 1H), 2.91 (s, 2H), 2.39 (s, 2H), 1.10 (s, 6H). ^13^C-NMR (101 MHz, DMSO) δ = 192.96, 167.26, 150.49, 148.25, 132.31, 129.63, 128.25, 123.57, 121.35, 120.12, 104.63, 51.32, 36.35, 34.92, 27.91. IR (KBr): λ = 3100, 2952, 2917, 2869, 1674, 1569, 1535, 1443, 1339, 1226, 830, 743, 630 cm^−1^.

2-(4-fluorophenyl)-6,6-dimethyl-6,7-dihydrobenzofuran-4(5H)-one (**15**-**33**)

White crystals, yield: 71%, m.p.: 118 °C. ^1^H-NMR (400 MHz, DMSO) δ = 7.81–7.76 (m, 2H), 7.30–7.25 (m, 2H), 7.15 (s, 1H), 2.87 (s, 2H), 2.36 (s, 2H), 1.09 (s, 6H). ^13^C-NMR (101 MHz, DMSO) δ = 193.07, 166.09, 163.01, 160.57, 152.78, 125.93, 121.11, 116.08, 115.86, 101.24, 51.35, 36.36, 34.92, 27.95. IR (KBr): λ = 3091, 2948, 2926, 2869, 1665, 1495, 1591, 1222, 1161, 1126, 635, 826 cm^−1^.

2-(2,4-difluorophenyl)-6,6-dimethyl-6,7-dihydrobenzofuran-4(5H)-one (**15**-**34**)

Colorless crystals, yield: 59%, m.p.: 155–158 °C. ^1^H-NMR (300 MHz, DMSO-d_6_) δ = 1.08 (s, 6H), 2.37 (s, 2H), 2.81 (s, 2H), 6.90 (d, 1H, *J* = 3.3 Hz), 7.20 (ddd, 1H, J = 3Hz), 7.40 (ddd, 1H, *J* = 2.7, 10.5 Hz), 7.82 (ddd, 1H, *J* = 6.6, 10 Hz). ^13^C-NMR (75 MHz, DMSO-d_6_) δ = 27.8, 34.8, 36.2, 51.2, 104.7, 104.9, 112.2, 114.1, 120.9, 127.4, 147.0, 156.3, 159.8, 163.3, 166.1, 192.8. EI-MS: *m*/*z* (%): 276 (M+, 71), 220 (66), 192 (60), 164 (10), 141 (100), 113 (11), 63 (4), 41 (3). IR (KBr): λ = 3066, 2956, 2904, 2871, 1648, 1513, 1465, 1413, 1224, 1120, 1045, 863, 794 cm^−1^.

6-6-dimethyl-2-(4-nitrophenyl)-6,7-dihydrobenzofuran-4(5H)-one (**15**-**41**)

Light yellow crystals, yield: 42%, m.p.: 180 °C. ^1^H-NMR (400 MHz, DMSO) δ = 8.30–8.26 (m, 2H), 8.03–7.98 (m, 2H), 7.55 (s, 1H), 2.93 (s, 2H), 2.40 (s, 2H), 1.10 (s, 6H). ^13^C-NMR (101 MHz, DMSO) δ = 192.96, 167.83, 151.56, 146.34, 135.18, 124.41, 121.53, 105.74, 51.33, 36.40, 34.90, 30.68, 27.92. IR (KBr): λ = 3101, 2960, 2960, 2929, 2866, 1672, 1574, 1535, 1444, 1338, 1223, 1128, 1041, 835 cm^−1^.

### 3.4. Antifungal Activity

Compounds **14** and **15** were evaluated for antifungal activity in vitro using commercial antifungals such as fluconazole for yeasts and itraconazole for fungi. Activities are reported in the minimum inhibitory concentration (MIC) values using the serial microdilution method on 96-well plates. The following reference strains were purchased from the American Type Culture Collection (ATCC): *Candida albicans* ATCC 24433, *C. glabrata* ATCC 66032, *C. guilliermondii* ATCC 6260, *C. krusei* ATCC 14243, *C. tropicalis* ATCC 750, *A. niger* ATCC 16404 and *A. fumigatus* ATCC 204305. The MIC values were determined according to the M27-S4 method of the Clinical Laboratory Standards Institute for yeasts [34] and the M38 method for fungi [35]. The RPMI-1740 (Sigma) culture medium buffered with 0.165 M of HEPES (Sigma) was used. The MIC values were defined as the minimum concentration that inhibits growth of ≥ 50% *Candida* spp. and % *Aspergillus* spp. in comparison to the controls. The compounds were dissolved in ethanol and serially diluted in the growth medium. The yeasts were incubated to 35–37 °C and the MIC values were measured at 24 and 48 h; the fungi were incubated to 35–37 °C and measured at 48 and 72 h.

## 4. Conclusions

The statement of Wu et al. that aromaticity may have considerable implications for molecular design was applied to obtain compounds with antifungal activity. The antifungal activity of two series of indol-4-ones was related to the aromaticity of the five-membered ring (evaluated with the NICS(0) and NICS(1) of the pyrrole ring) and the enolate and keto resonance structures; when aromaticity increased, the antifungal activity decreased for series I and increased for series II. A bioisosteric change [36,37] in indol-4-ones was made substituting the nitrogen atom with an oxygen atom in the five-membered ring to obtain benzofuran-4-ones. Ten benzofuran-4-ones were synthesized and tested against eight human pathogenic fungi. The equations obtained through simple linear regression using the NICS values and the antifungal activity predicted the experimental biological activity of benzofuran-4-ones; the biological activity calculated was statistically the same as the one obtained experimentally. Benzofuran-4-ones had a higher antifungal activity in filamentous fungi than indol-4-ones. The highest activity was for *A. niger* (1.95 µg·mL) which is around 128 times higher than the activity observed with indol-4-ones. The results support the statement of Wu et al. and show the importance of aromaticity in the molecular design of compounds with antifungal activity. The advantages of the use of NICS(0) and NICS(1) as criteria of aromaticity are that they are easy and economical to compute and at least in this model proposed provide reliable results. We are working on the study of the interactions between indole-4-ones and benzofuran-4-ones with their biological targets.

## Data Availability

The data presented in this study are available in the Appendix A.

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
