# Peer review of "Nucleus-Independent Chemical Shift (NICS) as a Criterion for the Design of New Antifungal Benzofuranones"

_molecules, 2021, doi:10.3390/molecules26165078_

Round 1

Reviewer 1 Report

Comments on the manuscript ID1342081 entitled “Nucleus independent chemical shift (NICS) as a criterion for the design of new antifungal benzofuranones”:

The study is a continuation of previous investigations of the authors (refs. 22 and 23) on the reactivity of indol-4-ones (supposed to inhibit lanosterol 14-α-demethylase (CYP51) via non-covalent ligand interactions) and their antifungal activity. In these previous studies, they correlate the electronic properties (hardness, electronegativity, electrophilicity, etc.) of the compounds with their antifungal activity. In the study herein, the authors describe a correlation between NICS (nucleus-independent chemical shifts) and antifungal activity of two series of compounds (indol-4-ones (series I: compounds 6 and 7a-7g; series II: 8a-8g). Aromaticity is found to have an opposite effect on the antifungal activity of series I and II members. New compounds (44 benzofuran-4-ones: compounds 14, 15-1 to 15-43) are modelled by substituting the indol-4-one nitrogen atom to oxygen atom and some of the proposed benzofuran-4-ones are synthesized to corroborate the theoretical predictions. The potential mechanism(s) of action of indol-4-ones and benzofuran-4-ones against pathogenic fungi is not discussed.

The importance of aromaticity to describe the interactions of organic molecules is unquestionable. NICS is really a simple and efficient aromaticity probe, that has been used extensively for the identification of aromaticity properties of molecules, ions, intermediates, and transition states since its introduction by Schleyer et al. [Paul von Rague ́ Schleyer,* Christoph Maerker, Alk Dransfeld, Haijun Jiao, and Nicolaas J. R. van Eikema Hommes, J. Am. Chem. Soc. 1996, 118, 6317].

The manuscript requires some improvement. Some shortcomings are listed below:

  • Title: should accurately describe the contents of the manuscript;
  • Abstract/introduction/conclusions sections should be modified: Wu et al. investigated hydrogen bonding–aromaticity relationships for different H-bonded substrates and concluded that “such relationships may have considerable implications for molecular design, as the functions of many heterocyclic biomolecules and drugs rely on the binding via H-bonds of a dominant keto/imine or enol/amine tautomer”, but this do not imply any NICS - antifungal activity relationship. Yes, “aromaticity may have considerable implications for molecular design” because of the importance of H-bonding for the interactions with biomolecules. H-bonding interactions of the studied indol-4-ones and benzofuran-4-ones with their biological target are not discussed in the manuscript, thus the emphasis on this article is unnecessary; the authors should consider referring to papers on the structure (aromaticity) - (biological) activity relationship.
  • Optimized geometries at least of the main compounds should be presented in the manuscript;
  • A figure with ghost atoms locations to calculate NICS – also;
  • The choice of the experimental and computational techniques should be motivated.

Reviewer 2 Report

The article demonstrates the novelty of the scientific approach and is of interest both in theoretical and practical terms. The results obtained support the use of nucleus-independent chemical shifts in the molecular design of compounds with antifungal activity.

There are minor typos especially in the names of compounds in the experimental part, e,g. "ciclohexane". In my opinion, I would remove the word "synthesis" before each compound in the experimental part indicating only the name of compounds, e.g., 5,5-Dimethyl-2-(2-oxo-2-phenylethyl) cyclohexane-1,3-dione (13-1) instead of "Synthesis of 5,5-dimethyl-2-(2-oxo-2-phenilethyl) ciclohexane-1,3-dione (13-1)". I recommend authors to correct the text of the experimental part for typos especially in the names of compounds. After eliminating these minor flaws the article deserves publication in Molecules.
